# Get Spatial from Non-Spatial Information: Inferring Spatial Information from Textual Descriptions by Conceptual Spaces

Omid Reza Abbasi [1], Ali Asghar Alesheikh [1,*] and Seyed Vahid Razavi-Termeh [2]

1 Department of Geospatial Information Systems, K. N. Toosi University of Technology, Tehran 19697, Iran; oabbasi@mail.kntu.ac.ir
2 Department of Computer Science & Engineering and Convergence Engineering for Intelligent Drone, XR Research Center, Sejong University, Seoul 05006, Republic of Korea; razavi@sejong.ac.kr
* Correspondence: alesheikh@kntu.ac.ir

**Abstract:** With the rapid growth of social media, textual content is increasingly growing. Unstructured texts are a rich source of latent spatial information. Extracting such information is useful in query processing, geographical information retrieval (GIR), and recommender systems. In this paper, we propose a novel approach to infer spatial information from salient features of non-spatial nature in text corpora. We propose two methods, namely DCS and RCS, to represent place-based concepts. In addition, two measures, namely the Shannon entropy and the Moran's I, are proposed to calculate the degree of geo-indicativeness of terms in texts. The methodology is compared with a Latent Dirichlet Allocation (LDA) approach to estimate the accuracy improvement. We evaluated the methods on a dataset of rental property advertisements in Iran and a dataset of Persian Wikipedia articles. The results show that our proposed approach enhances the relative accuracy of predictions by about 10% in case of the renting advertisements and by 13% in case of the Wikipedia articles. The average distance error is about 13.3 km for the advertisements and 10.3 km for the Wikipedia articles, making the method suitable to infer the general region of the city in which a property is located. The proposed methodology is promising for inferring spatial knowledge from textual content that lacks spatial terms.

**Keywords:** textual content; geo-indicativeness; conceptual spaces; topic modeling

**MSC:** 68T50; 68T30

## 1. Introduction

Geospatial information services are typically based on collected quantitative data about objects and phenomena. They can provide answers, limited by the accuracy of the underlying data, to spatial queries such as 'where is . . . ?' The answer is extracted from the stored data, which is basically coordinate-based. These services have been extremely helpful in a broad spectrum of disciplines ranging from agriculture [1], ecology [2,3], mining [4,5], and archeology [6] to urban planning [7,8], public health [9], and water resources [10]. The role of the rapid growth of spatial data acquisition technologies in the success of GIS is undeniable. Mappers all over the world are using equipment such as surveying tools, GPS receivers, and drones to provide accurate data for organizations and corporations. Clearly, this is a costly and time-consuming procedure. In addition, it requires a great number of people to be involved in the procedure.

Due to the developments of the internet and the high penetration rate of social networks in people's daily lives, a vast resource of information has been enabled. Of this, a noticeable share is enriched with spatial information. This information is mainly latent in textual content in the form of online books, user reviews, and blog posts [11]. As users utilize a natural language to communicate through the internet, this information appears in terms of qualitative descriptions of objects and events [12]. Although these descriptions

rarely contain the spatial location of objects (e.g., in terms of postal addresses [13]), they may contain place names, spatial relations, and directions. Research on the extraction of spatial relations from textual content and natural languages has been growing rapidly in recent years [14–16]. Such studies aim at extracting spatial information in three forms that correspondingly answer three types of queries. Specifically, they intend to find topological states, proximity relations, and spatial directions. Topological relations describe the way objects are connected. It is very useful in geographic information retrieval (GIR) and qualitative reasoning (QR) to know systematically how places noted in a text are located in relation to each other [17,18]. These relations often appear in natural language as prepositions [19]. While they inform us about the way objects are contacting each other, they do not compare the situation where the objects are in one topological form. For instance, if two objects are topologically disjoint, the topological models (e.g., RCC-8 [20]) do not provide information about how much the two objects are disjoint. Proximity queries aim to find objects near a reference object represented by a place name in text. As the exact locations of objects in textual corpora are not known, the interpretation of how much of a distance is considered as near is rather subjective [21]. Directions, whether cardinal or relative, and orientations are important spatial relations usually observable in textual content. Again, they may be subjectively interpreted and are rarely implemented in GIS [15].

While the focus of the existing literature on the subject has been on the extraction of the mentioned spatial information from spatial descriptors within texts, we argue that some spatial information can be inferred from non-spatial terms. As only a few sentences in textual resources contain spatial descriptors [22], the extraction of spatial information from non-spatial terms would be of high interest in applications that need to locate textual resources. Our approach is especially useful in applications where resources, rather than toponyms within resources, are to be located. To give an example, consider the description '*The house is furnished, has a roof garden, and a good view of the mountains is available*' for a house in the city of Tehran that has been advertised on the web. In this description, there is no sign of explicit spatial descriptors. However, when seeing the description, a resident of the city may learn that the house is located in a northern neighborhood. The resident knows that some aspects of housing in northern Tehran are different from those in other neighborhoods. In addition, there may be some terms in the description that do describe the house but imply the region in which it is located. In the example above, a roof garden is a feature of houses mainly located in northern Tehran. Also, the houses located in northern Tehran often have a clear view of the mountains.

Stock, et al. [19] categorized expressions within texts into three classes: geospatial, other-spatial, and non-spatial terms. The geospatial class includes those expressions that contain a geographical reference and a spatial relation term, hence referring to an absolute geographic coordinate. The other-spatial category refers to expressions that contain spatial relations but lack a geographic reference, leading to relative coordinates. Non-spatial expressions are those terms that do not fall within the previous categories. In this paper, we further extend Stock's non-spatial class into two distinct classes. In fact, we recognize another class of expressions that, while lacking spatial relations, contain terms implying geospatial information. We borrow the term geo-indicative from Adams and Janowicz [23] to refer to this class. The difference between the two classes is neither trivial nor explicit, as they both do not contain geospatial or other-spatial terms. However, geo-indicative expressions contain words that describe some characteristics belonging to a certain geographical region. This is in contrast with the non-spatial expressions, where terms describe features that do not belong to a specific region in the study area. Therefore, some terms that are identified as geo-indicative in some study areas may be non-spatial in other regions. While spatial relations appear in texts often as prepositional phrases, geo-indicative terms are latent in texts as nouns and adjectives. Table 1 shows real-world examples for each class of expressions.

**Table 1.** Different classes of expressions, their definitions, and real-world examples from our dataset.

| Class of Expression | Definition | Example [1] |
|---|---|---|
| Geospatial expressions | Contain direct references to geographical features, such as place names | The house is located on the south side of **the Valiasr metro station**. |
| Other-spatial expressions | Contain terms that refer to the relative position of features, such as topological relations | A supermarket is available **across** the street. |
| Geo-indicative expressions | Contain terms that do not directly refer to geographical features but describe the characteristics of a certain geographical region | The house is furnished, has **a roof garden**, and has a good view of **the mountains**. |
| Non-spatial expressions | Contain terms that neither refer to nor describe any specific geographical region | The house has two rooms. The building has been recently renovated, and the walls are wallpapered. |

[1] Words that can be used to infer spatial information from are bold.

In this paper, we aim to extract spatial information from the non-spatial terms that imply a space (i.e., geo-indicative terms) in large corpora with the help of the theory of conceptual spaces. The proposed methods prove more versatile than other approaches reliant on geospatial terms in texts, considering that few texts contain such terms. The method finds broad applicability in cases where a system is to be designed to georeference texts without explicit geospatial terms and conduct spatial analysis or recommend textual items to users. For instance, in the Divar application, from which we extracted one of our datasets, rental property advertisers have the option to provide exact location information by placing a pin on a map. However, if a user chooses not to specify the exact location of the property, the neighborhood in which the property is situated must be manually determined via a search box. In this real-world scenario, our proposed method could offer two valuable services: Firstly, the platform can utilize our approach to automatically predict and suggest the neighborhood name for the advertiser in advance, streamlining the process. Secondly, the platform's recommendation engine presently suggests items in other areas based on the adjacency of the neighborhoods to the current neighborhood. With our proposed method, the recommendation system gains greater flexibility since our estimation approach does not rely on the physical topology of neighborhoods. As a result, the recommendations can be more relevant and encompass a wider range of options, enhancing user experience and satisfaction. The contribution of our study is three-fold:

1. We propose two novel methods, specifically the regional conceptual space (RCS) and an adapted version of directional conceptual space (DCS) from [24]. These methods are founded on the conceptual spaces theory and are designed to represent terms in textual content within a high-dimensional space.
2. We suggest utilizing Moran's I as a measure for assessing the geo-indicativeness of terms in textual content. As far as we are aware, Moran's I has not been employed previously for the specific purpose of gauging the geo-indicativeness of terms in text.
3. We contrast the methods we propose with a previously suggested approach relying on Latent Dirichlet Allocation (LDA).

In Section 2, we review some studies conducted on the topic of spatial information extraction from textual content. In Section 3, the theory of conceptual spaces is introduced, and our adaptation of the theory is explained. In Section 4, the methodology of the work is presented. Then, we provide the results and discuss them in Section 5. Finally, we conclude the paper and highlight some suggestions for future research.

## 2. Related Work

In this section, we provide a review of research conducted on extracting spatial information from textual content. In this area of research, most studies have focused on identifying spatial relations, including topological relations, in texts. In other words, a significant share of such studies have chosen geospatial and other-spatial expressions to

extract spatial information from corpora. Since the focus of our study is on the extraction of spatial information from geo-indicative expressions, we limit our review to this topic.

Most studies in the area of the extraction of spatial information from text have utilized both geospatial and place names to estimate the location of resources. In almost all of these efforts, no distinction is made between place names and geo-indicative expressions, and both types are included in the computations. For instance, Han, et al. [25] focused on the feature selection procedure to enhance the accuracy of georeferencing. Their idea was that many words in texts are not useful (i.e., they are non-spatial). Hence, they proposed three feature selection methods to distinguish between terms that highly imply spatial information and non-spatial terms. They developed term frequency-inverse corpus frequency (TF-ICF), analogous to TF-IDF, along with information gain ratio (IGR) and maximum entropy (ME) as criteria to select features. The methods were applied on large geographical scales. A grid of $0.5'$ in $0.5'$ divided the Earth, and the tweets far from cities were eliminated. They reported a 10% improvement in the accuracy of georeferencing using only geospatial and geo-indicative words in comparison with using full features. They also noted that place names played a noticeable role in the improvement.

Place name recognition is vital to the georeferencing and enrichment of geospatial gazetteers. However, it is a time-consuming task and needs massive labeled datasets [26]. In most cases, there is even no place name within the text. It is especially true for short texts such as messages on social media and advertisements across the web. A third group of researchers focused only on the geo-indicative terms. For instance, Hollenstein and Purves [27] explored the terms used by users to describe city centers. They used geo-tagged Flickr photos to examine how different city centers across the globe are described. Then, the terms were used to derive the boundaries of the city centers. Wang, et al. [28] employed content analysis to explore the perception of green spaces in Beijing, China. They developed a structured lexicon based on which landscape features were extracted and fed into machine learning techniques. Their approach relies on prior knowledge about the phenomenon under study. Chang, et al. [29] proposed a probabilistic framework to estimate the city-level location of Twitter users based on the content of their tweets. In their approach, a classification component is used to automatically identify words with a high probability of referring to a specific city (the word *rockets* probably refers to *Houston*). Their results showed that 51% of Twitter users can be placed within 100 miles of their actual location. McKenzie and Janowicz [30] explored the regional variation between Foursquare's POI types. They applied a statistical hypothesis test to the dataset and found that there is regional variation between POI types that is not the result of random fluctuations. Adams and Janowicz [23] chose georeferenced Wikipedia articles and travel blog posts to train a Latent Dirichlet Allocation (LDA) model. They estimated a probability surface over the Earth for each topic. A Kernel Density Estimation (KDE) analysis was used to represent the probability of correspondence of each topic to the regions. They applied the method to places in the United States and achieved an accuracy of about 500 km for 75% of the test Wikipedia articles. In [31], the same authors extended their work and viewed the subject from a *semantic signature* perspective. Analogous to spectral signatures, semantic signatures can be used to illustrate various thematic features of the space. They showed that their proposed approach automatically categorizes documents into place types in existing ontologies. The focus of the mentioned study has been on the enrichment of knowledge graphs and inferring the relationships between place types in ontologies. While the topic of their study is closely related to ours, we proposed the use of conceptual spaces, rather than LDA, to build a similarity space of unstructured text. Therefore, the proposed method is more suitable in applications such as recommender systems, where similarity is at the core of computations.

The aim of the latter studies is similar to ours. They use only geo-indicative terms to infer the location of textual resources. In this article, we seek to extract geo-indicative expressions. It should be noted that geo-indicative terms refer to a specific region of the study area. Hence, the spatial distribution of documents containing geo-indicative

terms would be clustered. Therefore, geo-indicative terms inherently play the role of place names in text. Our study employs the theory of conceptual spaces, which, as a cognitive knowledge representation framework, is more suitable to the task of analyzing semantic information in textual corpora. In addition, as we construct a similarity space in which more semantically similar documents are placed closer together, our methodology becomes more interpretable and the properties of the space can be discussed. Another feature of our research is the size of the study area. In this study, we estimate the location of textual resources within the city of Tehran, the capital of Iran. According to the definition of geo-indicativeness, the different parts of the underlying study area must have distinguishing properties so that the language can describe them and discriminate from other regions. The previous studies mainly applied their method to large-scale areas such as the world and countries. Obviously, different regions of a country have specific properties, such as the accent, culture, geography, and climate, among many others, and can be described by diverse and distinguishing terms. We applied our method to the level of a city, which makes the prediction task more challenging. Unlike some other studies which seek to find the relationships between the location of a text and the user, our proposed method aims to infer the location of general textual resources, irrespective of the location of text's creator. Our approach does not inherently violate user privacy. Of course, if the method is to be applied in scenarios where privacy concerns matter, the users' consent is required. In the forthcoming sections, we present and scientifically compare our results with those of the above studies.

## 3. Materials and Methods

Figure 1 demonstrates the workflow of our study. The details of the methods shown in the figure are described in this section. First, the LDA method, as the reference method with which we compare our results, is explained. In order to provide the readers an insight into how the words and topics are represented over the study area, Kernel Density Estimation (KDE) is introduced. Then, the theory of conceptual spaces, as proposed by Gärdenfors [32] and formalized by Adams and Raubal [33], is briefly highlighted. After introducing the original theory, the modified versions that we utilized in the study is introduced. We propose two data-driven adaptations of conceptual spaces. The results of both methods are evaluated and compared against the LDA and a random estimator.

### 3.1. Semantic Signatures

Adams and Janowicz [23] proposed semantic signatures to address the problem of discovering latent spatial information in natural language. The method is based on Latent Dirichlet Allocation (LDA), an unsupervised probabilistic method over discrete data such as text corpora [34,35]. LDA is commonly used in the tasks of topic modeling [36] and recommendation [37]. It assumes that documents are probability distributions of topics, and each topic is a probability distribution of words within documents. Formally, it is assumed that a text corpus is comprised of $k$ topics $z_n$ and contains $|V|$ words. For a document $D$ containing $N$ words, the probability of assigning each topic to document $D$ is computed as [35]:

$$p(\mathbf{w}|\alpha, \beta) = \int_{\theta} p(\theta|\alpha) \left( \prod_{n=1}^{N} \sum_{z_n=1}^{k} p(w_n|z_n, \beta) p(z_n|\theta) \right) d\theta \tag{1}$$

where $\mathbf{w}$ is the set of words in document $D$, and $\theta$ is a $k$-dimensional random variable sampled from a Dirichlet distribution, and $\alpha$ and $\beta$ are the parameters of the model.

In order to represent the probabilities provided by LDA as surfaces over the study area, Adams and Janowicz [23] suggested using Kernel Density Estimation (KDE). Suppose

$\{X_1, \ldots, X_n\} \in R^d$ be an independent, identically distributed (iid) random sample from an unknown probability distribution $P$. KDE aims at estimating $P$ by [38]:

$$\hat{p}(x) = \frac{1}{nh^d} \sum_{i=1}^{n} K\left(\frac{x - X_i}{h}\right) \qquad (2)$$

where $K$ is a kernel function and $h \in \mathbb{R}^+$ is called bandwidth. KDE assigns a probability value to each sample. Then, the values are summed to yield a density estimator. The form of the distribution depends on the kernel function used in the equation. Two commonly employed kernels are Gaussian and spherical kernels. A Gaussian kernel is defined as [39]:

$$K(x) = \frac{exp\left(-\frac{\|x\|^2}{2}\right)}{\int exp\left(-\frac{\|x\|^2}{2}\right)dx} \qquad (3)$$

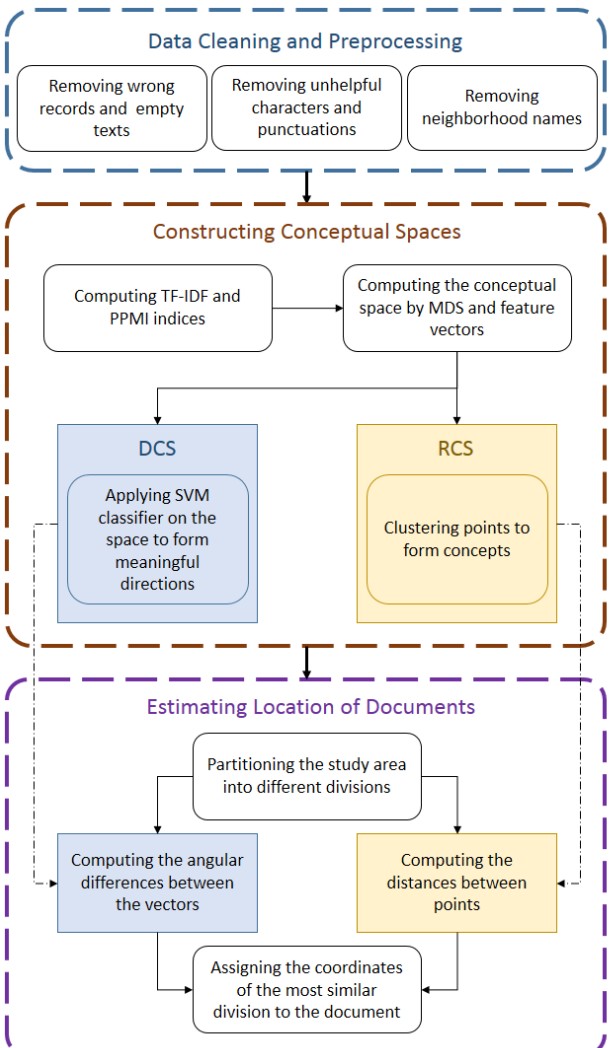

**Figure 1.** The workflow of the proposed methodology.

### 3.2. Conceptual Spaces

The theory of conceptual spaces [40] is a knowledge representation method [32] that uses geometrical structures to represent information, allowing for the definition of similarity relations between concepts [41]. This approach is distinct from associationist and symbolic levels of knowledge representation. Associationist models focus on connections among the

many parts of a system, while symbolic models manipulate symbols using rules written in First-Order Logic (FOL). Symbolic models typically process information sequentially, one symbol or rule at a time. This is due to the fact that symbolic representations are discrete and well-defined, and each symbol or rule can be processed independently of the others [42]. In contrast, associationist models often process information in parallel, with multiple connections between concepts being activated simultaneously [43]. This is due to the fact that associationist representations are distributed and overlapping, and the activation of one concept can lead to the activation of many others.

The theory of conceptual spaces tries to represent information based on geometrical structures. A significant advantage of this kind of representation is the ability to define similarity relations on information, and to relate the concepts topologically [44]. Moreover, inductive reasoning can be studied more conveniently at the conceptual level [45]. The geometrical forms of concepts can be used in both the explanatory tasks (for example, see Poth [46]), in which a cognitive theory is formulated and verified by means of experiments and observations, and the constructive tasks (for example, see Pol, et al. [47]), in which an artificial agent or system is designed to solve a specific problem. In this article, the focus is on the latter, as we aim to utilize the conceptual spaces to solve the task of inferring spatial information rather than to fathom the underlying complexities of human thinking regarding places.

A conceptual space is a similarity space spanned by a set $D$ of quality dimensions. Each dimension $d \in D$ is an aspect by which an object can be identified. Some quality dimensions that are related to each other are integrated into domains. While the initial works on Gärdenfors' conceptual spaces mainly included explicitly defined quality dimensions, there has been some recent effort to infer the conceptual space in a data-driven approach [48]. In this paper, we construct the conceptual space using a data-driven manner. In what follows, we introduce the elements used in the theory of conceptual spaces. While in the literature there are other elements such as fuzzy concepts [49] or hierarchical conceptual spaces [50], we define only the core elements needed by a conceptual space for the sake of simplicity.

**Definition 1.** *A concept is defined as $\mathcal{C} = \langle \mathcal{R}, \mathcal{P} \rangle$, where $\mathcal{R}$ denotes a set of convex regions in the space. $\mathcal{P}$ represents a prototype of the concept in the space. The prototype instance of the concept is assumed to be a point in the space, theoretically located in the center of the convex region representing the concept.*

**Definition 2.** *A property is defined as the value of a concept in only one quality dimension. Properties can be understood as seeing a concept from only one aspect.*

**Definition 3.** *An instance $\iota$ is defined as a point within a conceptual space that represents an object in the real world. An instance is produced when a value is determined for all quality dimensions in the space.*

In this article, we construct the conceptual space in a data-driven manner and propose a modification in the representation of concepts.

### 3.3. Data-Driven Conceptual Spaces

As stated in the previous section, the theory of conceptual spaces represents concepts as convex regions in a metric space. In most cases, since there is no prior knowledge about the underlying phenomenon, the quality dimensions cannot be identified explicitly. Therefore, a data-driven approach is essential to building the conceptual space from the bottom up. In this article, we learn the conceptual space of our corpus using some techniques used in machine learning and Natural Language Processing (NLP).

Suppose we are going to represent a corpus $C$ containing documents $\{D_1, \ldots, D_N\}$. Each document is first tokenized into a Bag of Words (BOW) containing all of the words in the documents. Similar to any NLP task, and as some tasks need only specific parts of speech (POS), the BOWs are preprocessed, and the tokens that are not useful for the

specific task are removed. Then, each token in each BOW is assigned an index to provide the salience of the words in the documents. TF-IDF is a common index to this aim. Given a set $D$ of documents, TF-IDF scores the word $w$ in the document $d$ as [51]:

$$TF\_IDF(w, D, C) = TF(w, D) \times IDF(w, C) \tag{4}$$

where

$$TF(w, D) = \frac{c(w, D)}{|D|} \tag{5}$$

and

$$IDF(w, C) = log \frac{|C|}{|\{D \in C | w \in D\}|} \tag{6}$$

where $c(w, D)$ is the number of times a term $w$ occurs in the document $D$. Also, $|D|$ and $|C|$ denote the number of words in the document $D$ and the number of documents in the corpus $C$, respectively.

Positive point-wise mutual information (PPMI) is similar to TF-IDF in that it also scores the words in a document. Some researchers have shown that it works better than other scoring methods for the purpose of semantic similarity [46]. The index is calculated as:

$$PPMI(w, D) = max\left(0, log\left(\frac{p_{w,D}}{p_{w,*} \times p_{*,D}}\right)\right) \tag{7}$$

where

$$p_{w,D} = \frac{c(w, D)}{\sum_{w'} \sum_{D'} c(w', D')} \tag{8}$$

and

$$p_{w,*} = \sum_{D'} p_{w,D'} \tag{9}$$

and

$$p_{*,D} = \sum_{w'} p_{w',D} \tag{10}$$

By applying the scoring index to each document $D$, a vector $v_D$ is formed. This vector contains the index values of all words in the corpus. Since the index vectors will be very sparse, they are not suitable to be directly used in constructing the conceptual space. Hence, a Multi-Dimensional Scaling (MDS) technique [52] is applied to the vectors to transform them into a similarity space of documents. While MDS itself is not specific to conceptual spaces theory, it has been denoted by Gärdenfors [40] and others [53] as a method of constructing the conceptual spaces and identifying the quality dimensions. Given the pairwise similarities of a set of objects, MDS estimates their coordinates as points in a similarity space [52]. It has been widely used in cognitive sciences [53] and also in the theory of conceptual spaces [54].

TF-IDF and PPMI offer a clear and easily understandable representation of words based on their significance within a specific document. The assigned weights by TF-IDF and PPMI indicate how relevant a word is to a document relative to a larger corpus, which is particularly useful when analyzing the geo-indicativeness of words in our spatial inference approach. In contrast, more recent methods such as doc2vec employ neural networks for model training, which reduces their interpretability. It is essential to note that one advantage of conceptual spaces over connectionist methods such as neural networks is their interpretability. In addition, TF-IDF and PPMI are relatively simple and computationally efficient methods for representing textual data. In contrast, word embedding methods often involve more complex models and require extensive training on large corpora.

In order to represent concepts as regions in the resulting space, spatial clustering would classify similar documents, represented by points, into single classes. Finally, the convex hull of the points yields the desired regions. In this paper, the conceptual spaces provided by the above procedure are called *regional conceptual spaces* (RCS).

Although the original theory of conceptual space revolves around the prototype theory [41], in the case of inferring the conceptual space from data, it might be rare to have a prototypical instance that represents the ideal case of the category in the middle of each class. Therefore, we propose a modification to the way classes are represented. To this end, we partition the resulting similarity space for each word using a Support Vector Machine (SVM) classifier. This leads to two classes of points: a class representing documents that do not contain a given word (negative class), and another class representing documents that contain that word (positive class). Then, we calculate the equation of the hyper-plane classifier. The direction perpendicular to the hyper-plane demonstrates the direction in which the word plays a prominent role. The farther a specific document is placed from the hyper-plane in the direction of positive documents, the more salience the word has in the document. As this classification does not potentially yield perfect accuracy, we consider only those words for which the classification has promising accuracy. This is to ascertain that the directions found have strong meanings. However, even after constructing the hyper-plane only for those words with high accuracy of classification, there may remain a high number of directions depending on the number of words considered. In addition, some vectors may have similar meanings. As we are handling a similarity space, the vectors with similar meanings are pointing towards a close direction. The directions can be clustered to achieve more robust and meaningful directions within the constructed space. We call the result of this procedure *directional conceptual spaces* (*DCS*). Figure 2 compares the two approaches to constructing the data-driven conceptual space mentioned above. Abbasi and Alesheikh [24] presented a method similar to DCS with a focus on utilizing it in recommendation systems. They measure the similarity of items of a recommender system by calculating the difference between the lengths of vectors projected on meaningful directions. The DCS method used in the present study uses the angular differences between the vectors to calculate their closeness. They evaluated the results by real users of a recommender system. The main difference between the present study and [24] is that we propose both DCS and RCS and compare their performance against LDA a random estimator, without focusing on recommendation algorithms. In addition, we partitioned the study area into equally-sized cells to compensate for the problem of the density of points in KDE, which is discussed in the 'Results' section. Furthermore, we suggest utilizing Moran's I, along with the Shannon entropy, as a measure for assessing the geo-indicativeness of terms in textual content.

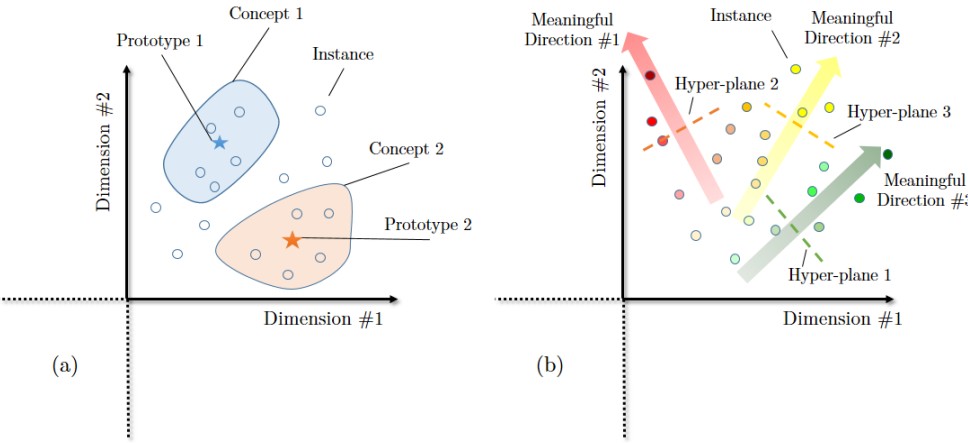

**Figure 2.** A schematic comparison of the two proposed versions of data-driven conceptual spaces (modified from [24]). (**a**) Regional Conceptual Spaces represent concepts as convex regions in a high-dimensional space; (**b**) Directional Conceptual Spaces represent concepts as directions in a high-dimensional space.

After constructing the conceptual space, the documents need to be located in the geographical space. To this aim, the study area is partitioned into different divisions. Then,

for a new document, the most appropriate division is determined based on the similarity of their associated concepts or their closeness to meaningful directions. That is, in the case of RCS, the division with which the document shares the most common regions in the conceptual space is selected as the estimated division, and its geographical centre is assigned to the document. In the case of DCS, the division with which the document has the least angular difference is selected as the estimated division.

## 4. Datasets

In this paper, we used a dataset of advertisements for rental properties and a dataset of Persian Wikipedia articles, both related to the city of Tehran, Iran. To collect the first dataset, we crawled the Divar website (https://divar.ir, accessed on 21 March 2023), a popular platform used in Iran to advertise properties for sale or rent. The dataset contains about 200,000 records. After removing records that do not contain the coordinates of the property or lack a textual description, 11,393 records of advertisements remained. Each record includes a title and a description of the property in Persian, pricing information, the construction year, the area of the property, the name of the neighborhood where it is located, and locational data including latitude and longitude. In our study, we only processed the descriptions related to each advertisement. In addition, all place names were removed from the descriptions. In order to process the descriptions, we utilized the Stanza library [55], a Python natural language processing package that is equipped with pre-trained models for Persian language. The second dataset was extracted from the Wikimedia Query Service by writing queries in SPARQL. It contains 704 Persian Wikipedia articles that are geolocated in Tehran, Iran. The articles are full-length and cover various topics such as cultural sites, monuments, parks and gardens, and events. In our study, we use 80% of the records for the training purpose, and employ the remaining records for testing. In Table 2, the datasets used in the study are listed, and an example of each dataset is provided.

**Table 2.** The description of the datasets used in the study.

| Dataset | No. of Records | Example [1] |
|---------|----------------|-------------|
| Divar | 11,393 | "220 m, 3 bedrooms, very well designed, large and spacious living room, fully furnished kitchen, southern exposure, very large meeting hall, the most cozy region, very stylish and luxurious and furnished lobby, the roof garden is very well equipped and wooded, lobby, and resident janitor; it has two parking lots, private pool, sauna, and Jacuzzi." |
| Persian Wikipedia | 704 | "The National Museum of Iran is located in Tehran, Iran. It is an institution formed of two complexes; the Museum of Ancient Iran and the Museum of Islamic Archaeology and Art of Iran, which were opened in 1937 and 1972, respectively. The institution hosts historical monuments dating back through preserved ancient and medieval Iranian antiquities, including pottery vessels, metal objects, textile remains, and some rare books and coins. It also includes a number of research departments, categorized by different historical periods and archaeological topics." |

[1] The examples are translated into English.

## 5. Results and Discussion

In this section, we present the results of applying the methods described in Section 3. Specifically, we apply our proposed approaches using conceptual spaces and compare the results with those of the LDA method.

In order to preprocess the textual descriptions, all of those properties for which the textual description is less than 10 words were removed. Considering that the length of feature vectors is equal to the number of all words in the corpus, this would help in compensating for the issue of feature vectors' sparsity. First, the descriptions were segmented into sentences, and then sentences were tokenized into words. To reduce the volume of data in processing, in this paper we only considered nouns and adjectives in sentences. On the other hand, by emphasizing nouns and adjectives, we aim to prioritize words that provide more explicit geo-indicative and descriptive context. We emphasize that the choice of parts of speech is dependent on the use case. In our study, verbs do not seem to be geo-indicative for rental property advertisements and Wikipedia articles about Tehran. However, the use of verbs might be better suited in scenarios where identifying functional regions is important. In addition, we considered only those words that were pointed out in the whole corpus at least 50 times. This choice is based on a heuristic threshold to filter out less frequent and potentially noise-inducing words and to focus on the most informative and discriminative words, capturing the key features and characteristics of the documents. Since data-driven conceptual spaces are very prone to the quality of the data, if the data is noisy or incomplete, then the conceptual space may be inaccurate or misleading. This ensures the strength of the meaning found in future steps and enhances the accuracy of classification. The same pre-processing procedure was used for both LDA and the proposed methods to ensure a fair comparison between them.

We selected the number of LDA topics $k$ through qualitative evaluation of topic interpretability on each dataset. The appropriate $k$ value depends on the semantic domain of the corpus. Since the rental advertisements cover a narrower domain than Wikipedia, their content reflects fewer meaningful topics. After testing different $k$ values, we chose $k = 6$ for the rental advertisements and $k = 10$ for the Wikipedia articles by judging which values produced the most coherent, interpretable topics. In Table 3, example topics from each dataset, both geo-indicative and non-geoindicative, are shown. After training the LDA model, we construct a probability surface using IDW to test whether a routine interpolation technique can suitably represent the probabilities. Figure 3 illustrates the geographical distribution of advertisements belonging to Topic #D1 in Table 3, which contains the words *pool*, *sauna*, *jacuzzi*, *conference_hall*, and *luxurious*. The surfaces are drawn based on the probabilities yielded by the LDA topic modeling.

**Table 3.** Some topics identified in each dataset along with their entropy values.

| Dataset | ID | Topic [1] | Entropy | Moran's I |
|---|---|:---:|:---:|:---:|
| Divar | #D1 | (*pool, sauna, Jacuzzi, conference_hall, luxurious*) | 4.52 | 0.344 |
| | #D2 | (*city_center, bazaar, store, accessibility, commercial*) | 5.67 | 0.472 |
| | #D3 | (*loan, under_construction, cooperation, phase, vacant*) | 10.60 | 0.022 |
| | #D4 | (*owner, room, alley, door, telephone*) | 33.02 | −0.471 |
| Persian Wikipedia | #W1 | (*museum, register, national, the_King, traditional*) | 7.51 | 0.453 |
| | #W2 | (*university, faculty, science, institute, research_center*) | 10.21 | 0.320 |
| | #W3 | (*station, line, metro, neighborhood, square*) | 19.55 | 0.154 |
| | #W4 | (*neighborhood, district, city, municipality, area*) | 29.02 | −0.745 |

[1] The words are translated into English.

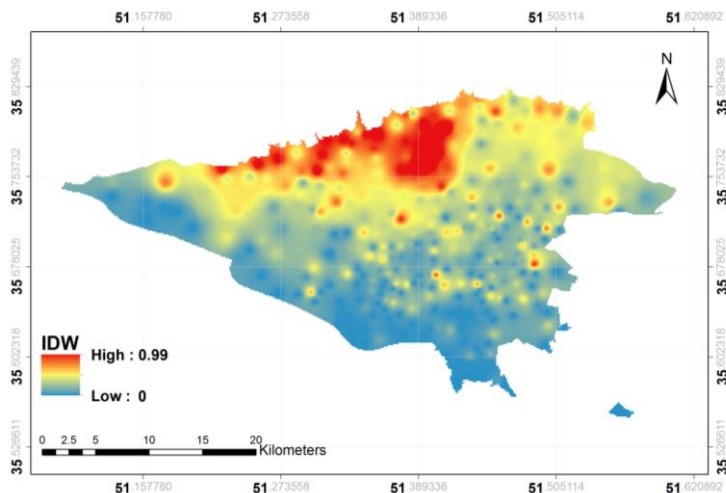

**Figure 3.** The probability surface calculated by IDW for Topic #D1 (refer to Table 3) found in the corpus.

As seen in the figure, the surface shows a probability distribution for this topic, which is strongly correlated with the northern areas of the city. However, the surface highly respects single points, and the map is too rough. We calculated KDE as another approach to representing the probability surface. As discussed in Section 3.1, the location of documents is involved in the calculations of KDE. Therefore, those areas with more advertisements would have higher probabilities. In order to compensate for this issue, following Adams and Janowicz [23], we partitioned the study area into different divisions. A grid comprising cells of size 1 km in 1 km each divided the city. Then, the averages of probabilities for each topic were calculated and assigned to the centroid of the cells. Figure 4 manifests the distribution of the same topic (Topic #D1) represented by KDE.

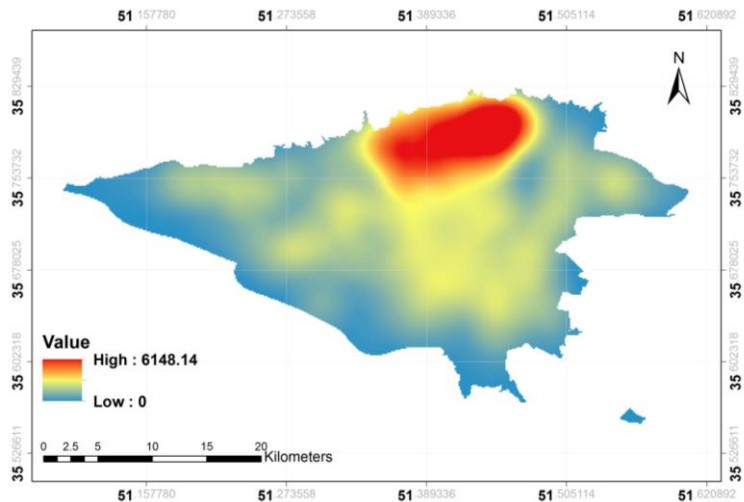

**Figure 4.** The probability surface calculated by KDE for Topic #D1.

The resulting surface provided by KDE is much smoother than that provided by IDW. This is due to the fact that, according to Equation (2), KDE averages the probability distributions over sample points, compensating the role of single points with high values. However, in IDW, single points can have significant effect on the resulting surface. Following Abbasi and Alesheikh [24], we further analyzed the topics found to measure the degree of geo-indicativeness of each topic. From one point of view, a topic is geo-indicative if it is associated with some neighborhoods while having low association probabilities with other neighborhoods. As a result, a geo-indicative topic would not be uniformly dispersed

through the study area. To measure the dispersion, we compute the Shannon entropy of the probabilities by [56]:

$$H(p) = -\sum_{1}^{N} p_i log(p_i) \tag{11}$$

Shannon entropy calculates how much the outcome of a probability distribution would be *surprising* [57]. The lower the entropy, the lower the amount of surprise. This is interestingly in accordance with the concept of geo-indicativeness. The terms of a geo-indicative topic have to, *unsurprisingly*, describe the region to which they belong. Table 3 lists some topics found and their associated entropies.

In addition to Shannon entropy, we propose the Moran's I index to measure the geo-indicativeness of topics. The Moran's I is a measure of spatial autocorrelation of data. It works based on both feature location and attributes. That is, it measures how much near features have similar attributes [58]. It is calculated as:

$$I = \frac{n}{S_0} \frac{\sum_{i=1}^{n} \sum_{j=1}^{n} w_{i,j} z_i z_j}{\sum_{i=1}^{n} z_i^2} \tag{12}$$

where $z_i$ is the deviation of an attribute for feature $i$ from its mean, and $w_{i,j}$ is the spatial weight between feature $i$ and $j$. $S_0$ is the aggregate of all of the spatial weights, and is calculated as:

$$S_0 = \sum_{i=1}^{n} \sum_{j=1}^{n} w_{i,j} \tag{13}$$

For a detailed explanation of computing Moran's I, the reader should refer to [59]. The value of Moran's I fluctuates between $-1$ and $1$, where a value of $-1$ indicates a strong dispersion of similar attributes, a value of 0 indicates a random spatial distribution of similar attributes, and a value of 1 indicates a strong clustering of similar attributes. We expect that geo-indicative topics should have a positive Moran's I. If the value of Moran's I is close to one, it is highly probable that the topic is geo-indicative, as it means the topic is spatially clustered. Conversely, if it is negative, the topic is not geo-indicative. If the Moran's I is close to zero, then the topic is probably not geo-indicative. We suggest that the two criteria, i.e., the Shannon entropy and the Moran's I, be considered simultaneously. This is especially important in the case that the Moran's I shows a borderline value for a topic.

The first three topics in the rental property advertisements listed in Table 3 show high degrees of geo-indicativeness (low entropy). Topic #D1 is the one illustrated in Figures 3 and 4, which refers to properties mostly located in northern neighborhoods of the city. Topic #D2 is vividly related to the central neighborhoods that are the commercial heart of the city and are known for their ease of accessibility. Topic #D3 refers to those properties that are under construction and are mostly located in the vicinity of the city. While the above topics clearly represent specific regions of the city, we could not interpret the rest of the topics as geo-indicative. For instance, Topic #D4 contains general features that can be seen in any neighborhood in the study area. This is in contrast with Topics #D1 and #D2 where their related words are associated with northern and central neighborhoods, respectively. In the case of the Wikipedia articles, the entropy values of topics are generally higher than those of the advertisements dataset. This is due to the fact that the themes of the Wikipedia articles are diverse, and many of them describe features that are not associated with their underlying space. For example, Topic #W2 refers to the articles about universities and research institutes. The universities and research institutions in Tehran are not located in a specific region. The entropy of Topic #W4 is more interesting. The topic contains articles that introduce neighborhoods of the city. Although the documents in this topic inherently describe their underlying space, they are, expectedly, distributed over the study area, leading to high entropy and negative Moran's I. Topic #W1 is related to museums and historical sites, such as palaces, which in turn relate mostly to central neighborhoods and

some of the northern neighborhoods. Figure 5 compares a geo-indicative topic (Topic #D2) with a non-geo-indicative topic (Topic #D4).

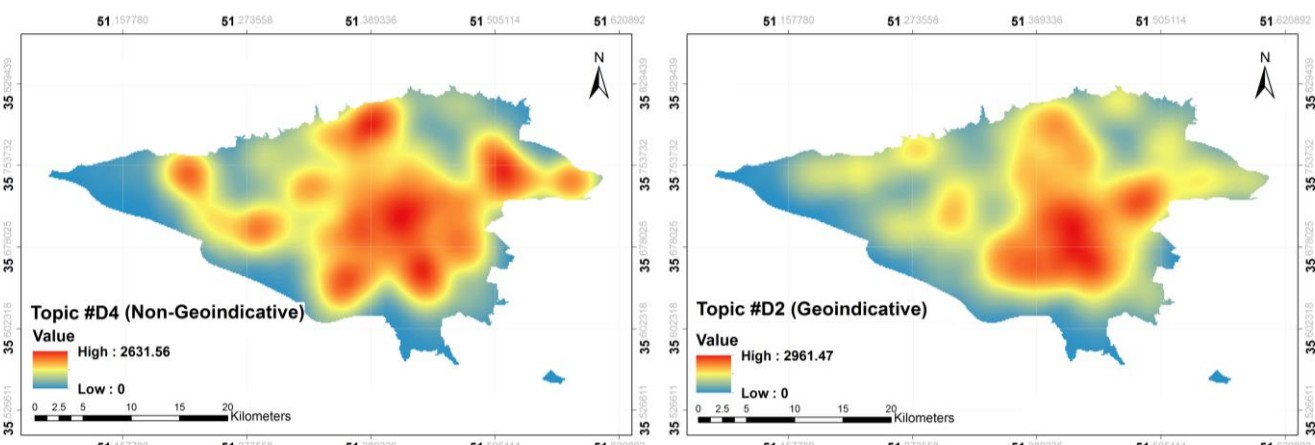

**Figure 5.** A comparison of the distribution of a geo-indicative (Topic #D2) versus a non-geo-indicative (Topic #D4) topic.

The right panel of Figure 5 verifies the centrality, explained above, associated with Topic #D2. However, for the left panel, the hotspots are almost evenly distributed over the study area, and a specific region cannot be easily adopted for the topic.

We also applied the proposed data-driven conceptual spaces methods (Section 3.3) to the same datasets. For this purpose, we considered only nouns and adjectives to reduce the volume of BOWs and enhance the speed of computations. In addition, all unnecessary tokens, such as pronouns and stop words, were removed from the BOWs. Both the TF-IDF and PPMI indices were calculated to score the salience of each word in the documents. For the purpose of implementing MDS, clustering the points in RCS, and applying SVM to partition the space in DCS, the scikit-learn package was employed. An issue with the DCS approach is that considering all words in the corpus and applying SVM to each is very time-consuming and computationally intensive. In addition, many words do not appear to have signs of being geo-indicative. Therefore, after computing TF-IDF and PPMI, we proceeded with only the top 10 percent of words in terms of these indices. Then, a dissimilarity matrix, as the input for MDS, was computed for the documents. Since we have no information about the dimensions of the resulting conceptual space, we computed spaces with 5, 10, and 20 dimensions. An SVM classifier with a linear kernel was trained to construct the DCS. The accuracy score of the classification was estimated, and those higher than 80 percent were considered as strong directions in the corpus. Table 4 shows the accuracy score of classification for 10 words also found by LDA and listed in Table 3 as Topic #D1 and Topic #W1.

In most cases, the PPMI score outperforms the TF-IDF, which suggests that it is a better scoring scheme for constructing directional conceptual spaces. Nevertheless, the classification for broad terms such as *luxurious* (Persian: شک) yields the worst results, regardless of the scoring index. The importance of the number of dimensions should not be overlooked. The pattern of convergence can be seen in the accuracies in most cases, and higher dimensions generally provide better classification accuracy. We applied a vector clustering algorithm to group the five directions of the words. Analogous to the prototypes in RCS, the farthest document from the classifier hyper-plane is identified as the most prominent feature for the word under consideration. In order to find the similarities among documents, we first projected the vectors onto the found direction and calculated the distance between those documents and the others in the found direction. Then, we sorted out the documents based on their normalized similarities. Figure 6 illustrates the location of

the most prominent document for the words *Jacuzzi* and *museum*, and their corresponding similarity surface.

**Table 4.** The accuracy score of SVM classification using TF-IDF and PPMI in 5, 10, and 20-dimensional conceptual spaces.

| Dataset | Word | TF-IDF | | | PPMI | | |
|---|---|---|---|---|---|---|---|
| | | **5D** | **10D** | **20D** | **5D** | **10D** | **20D** |
| Divar | *pool* (استخر) | 0.36 | 0.57 | 0.61 | 0.52 | 0.62 | 0.76 |
| | *sauna* (سونا) | 0.65 | 0.68 | 0.69 | 0.70 | 0.73 | 0.76 |
| | *Jacuzzi* (جکوزی) | 0.44 | 0.52 | 0.53 | 0.50 | 0.55 | 0.56 |
| | *conference_hall* (سالن اجتماعات) | 0.52 | 0.54 | 0.55 | 0.47 | 0.50 | 0.52 |
| | *luxurious* (شیک) | 0.24 | 0.41 | 0.54 | 0.27 | 0.41 | 0.55 |
| Persian Wikipedia | *museum* (موزه) | 0.65 | 0.71 | 0.75 | 0.65 | 0.72 | 0.75 |
| | *national* (ملّی) | 0.43 | 0.46 | 0.50 | 0.40 | 0.45 | 0.49 |
| | *register* (ثبت) | 0.49 | 0.52 | 0.54 | 0.51 | 0.55 | 0.58 |
| | *Shah* (شاه) | 0.53 | 0.56 | 0.56 | 0.55 | 0.56 | 0.56 |
| | *traditional* (سنتی) | 0.39 | 0.42 | 0.44 | 0.41 | 0.44 | 0.46 |

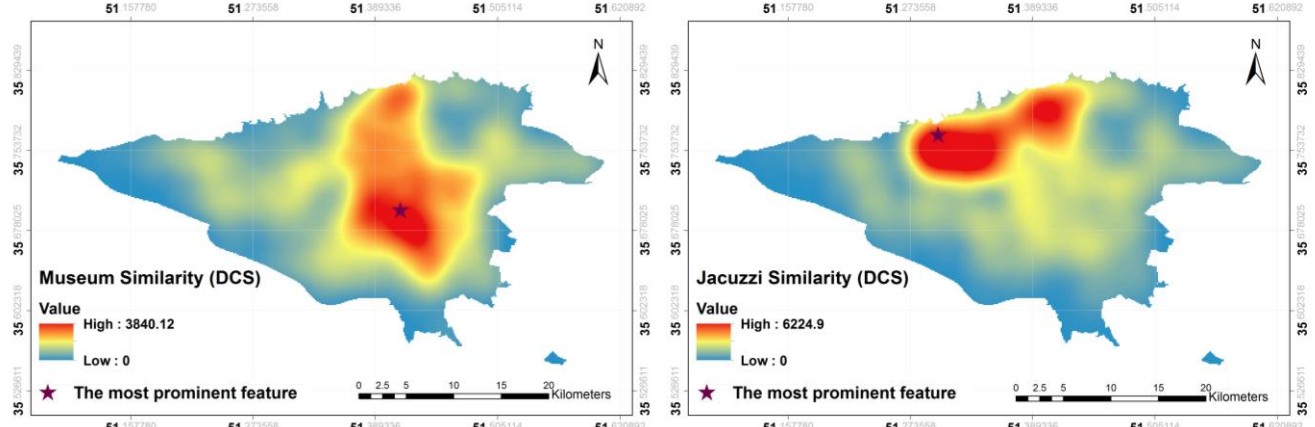

**Figure 6.** The locations of the most prominent documents containing the word *jacuzzi* in the Divar dataset and the word *museum* in the Wikipedia dataset and their similarity surface.

In the case of RCS, documents with similar words were grouped into one cluster. To group similar documents, we utilized *k*-means clustering. We tested *k* in the range of 1–30 clusters, running *k*-means with each value five times to account for variability. To select the best *k*, we evaluated each clustering using the Silhouette Coefficient metric, which measures how tightly grouped the data is within each cluster. The Silhouette Coefficient was highest with $k = 20$ clusters. In the case of rental property advertisements, out of 20 clusters, only two seemed geo-indicative. The results of RCS are shown in Figure 7, where the top panels illustrate the two geo-indicative clusters and the bottom panel shows a non-geo-indicative group.

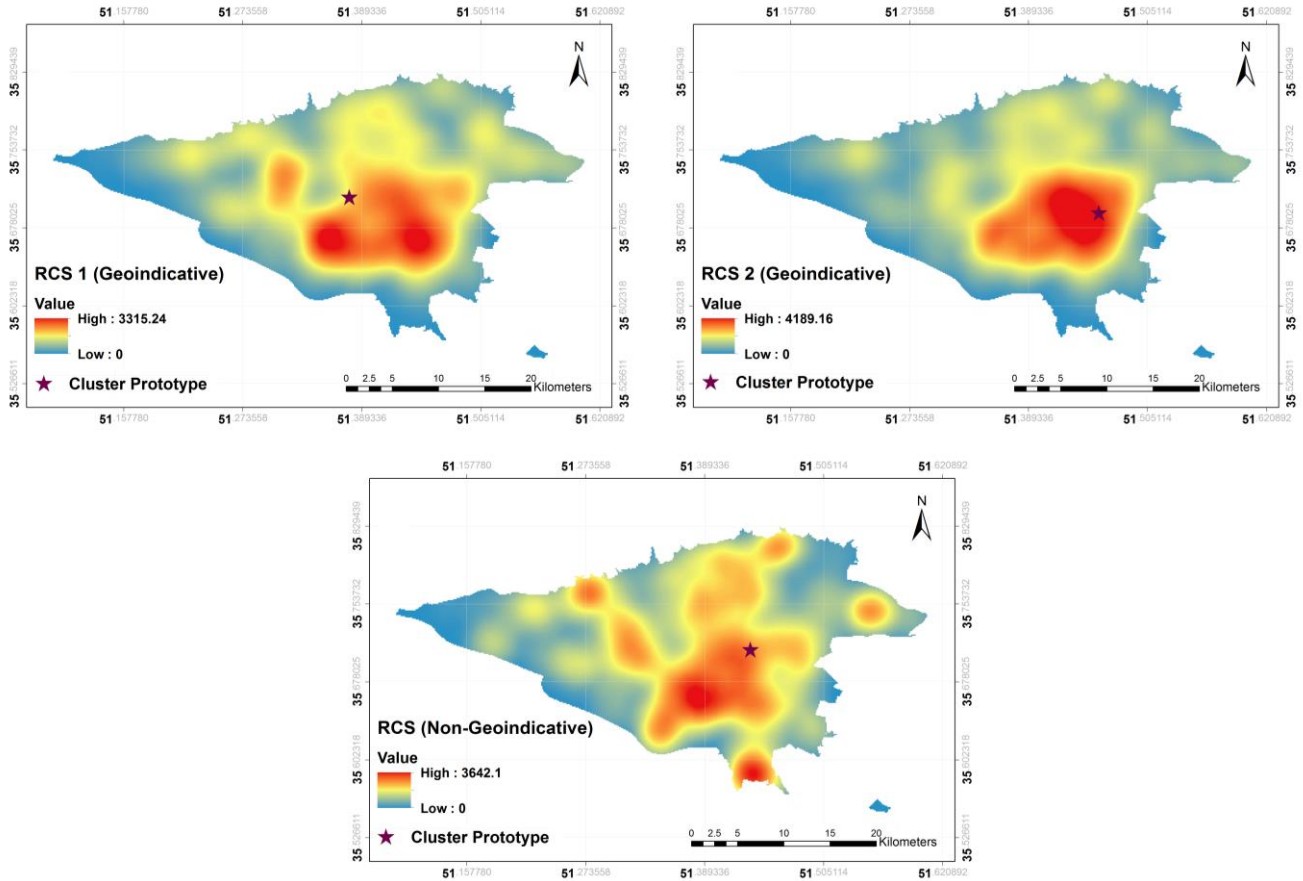

**Figure 7.** The probability surfaces generated by the RCS method for two geo-indicative and one non-geo-indicative regions in the case of rental property advertisements.

A disadvantage of RCS is that the meaning of clusters is not intuitive, as we have no interpretation of the dimensions or the clusters. Since the classification is done manually in DCS, the issue is more or less compensated in DCS. A suitable measure of the geo-indicativeness of the clusters in RCS is the distance between the prototype and the cluster mapped on the geographical space. As seen in Figure 7, although for the two geo-indicative clusters the prototype is located almost in the middle of the hot spots, the location of the prototype for the non-geo-indicative cluster is far from the hot spots. This means that while the documents in the semantic space have been close to each other, they are dispersed in the geographical space, pertaining to the fact that the semantics for this cluster do not imply a specific region in the geographical space. In order to compare the results of the three approaches, we computed the accuracy of the location prediction in terms of the distance between the predicted locations and the true locations for the test dataset (20 percent). Figure 8 plots the cumulative distance error for all methods and datasets.

The plot also shows the results of a random estimator as a benchmark for the prediction. All of the methods have better accuracies than the random estimator. However, the accuracy of RCS is worse than the other two in both datasets and, in some places, is even equal to the random prediction. DCS is the most accurate approach with an average distance error of 13.3 km for the Divar dataset and 10.3 km for the Wikipedia dataset. Since the size of the city is about 40 km by 20 km, this accuracy is suitable to predict the general region in which a property is located. That is, it can be inferred that a given document is located in the northern parts of Tehran. The worst prediction of DCS has been about 22.5 km, which is approximately equal to half of the length of the city. This value for RCS and LDA is about 31 km and 28 km, which shows that they completely fail in predicting some documents. DCS can promisingly predict the cardinal direction in which the property is located. It is

predicting the actual locations better than LDA by 10% and 13% on average, respectively, for the Divar dataset and the Wikipedia articles.

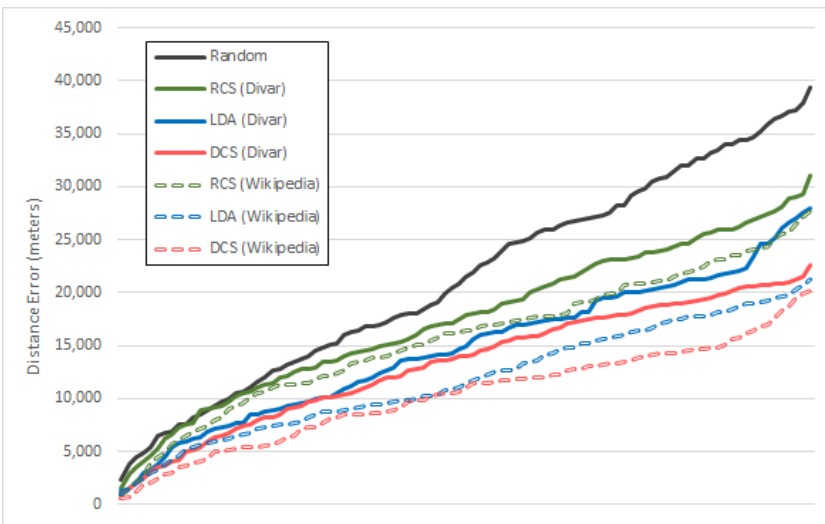

**Figure 8.** A comparison of the implemented methods in terms of cumulative distance error for both datasets.

## 6. Conclusions

In this paper, we propose a method to extract spatial information and estimate the location of textual content lacking place names or spatial relations. We argued that some terms within texts do not directly refer to places in geographical space but do imply spatial information. The focus of past studies on inferring spatial information from text has been on extracting spatial relations and place name identification and disambiguation. We proposed two different approaches to infer data-driven conceptual spaces (DCS and RCS) in an unsupervised manner. The proposed method was applied to a dataset of rental properties and a dataset of Persian Wikipedia articles, both for Tehran, Iran. We calculated the degree of geo-indicativeness of terms by their entropy. Then, we compared the results with those of LDA. The results showed that DCS outperformed LDA and RCS in terms of accuracy of prediction. By utilizing DCS, the location of properties can be predicted at 9.6 km, which is suitable for inferring the general region in which a property is located. Considering the volume of textual content lacking spatial information, this method can be a promising approach to inferring spatial information from texts.

The main advantage of our proposed method is that it is grounded in the theory of conceptual spaces, which provides a natural way to represent the meaning of words and their relationships in a geometric space. This means that the properties of the space can be discussed and interpreted in terms of the concepts that they represent. For example, we can visualize the relationships between different concepts in the space and see how they relate to each other. In contrast, LDA relies on statistical models that are not easily interpretable. While it can provide useful insights into the topics and themes that are present in a corpus, it can be challenging to understand how these topics relate to each other and to the underlying concepts that they represent. On the other hand, the proposed method offers a more flexible knowledge representation approach. Since documents are represented by points in conceptual spaces, they can easily be grouped into clusters and form concepts, which provides a multitude of analytical capabilities, such as the topological relationships between different concepts. Furthermore, since similarity computations are more intuitive in conceptual spaces, the proposed method is of great help in recommender systems.

One potential implication of our approach is that it can be used to analyze social media data and identify the geographic locations of social media users. This can be

useful for understanding the spatial distribution of social media users and their behavior patterns, which can be valuable information for marketing, social science research, and public policy. While our proposed method infers spatial information of texts, rather users, respecting individuals' privacy is a fundamental principle, and any location inference or data processing should adhere to applicable privacy laws and regulations. Furthermore, our approach is scalable and can handle large volumes of textual data efficiently. This makes it suitable for processing large datasets such as social media posts, news articles, historical documents, and online reviews. In terms of uses, our approach can be applied in various domains such as disaster management, tourism, and urban planning. For example, in tourism, it can be used to identify popular tourist destinations and understand the behavior patterns of tourists. In urban planning, it can be used to analyze social media data and understand the spatial distribution of urban activities, which can inform city planning decisions.

The identification of geo-indicative topics in our paper is based on interpreting the entropy and Moran's I values. This interpretation is manually carried out by comparing the entropies among different topics. As a suggestion, future studies can focus on defining a baseline by which a topic is considered geo-indicative. In addition, various operations can be defined over the conceptual spaces, which makes logical inferences possible. For example, the intersection of two concepts (i.e., regions or directions) may lead to the construction of a new meaningful concept. By studying such operations, the analytical capabilities of the proposed approaches can be improved.

**Author Contributions:** Conceptualization, O.R.A.; Data curation, O.R.A.; Formal analysis, O.R.A.; Funding acquisition, S.V.R.-T.; Investigation, O.R.A. and A.A.A.; Methodology, O.R.A.; Project administration, A.A.A.; Resources, S.V.R.-T.; Software, O.R.A.; Supervision, A.A.A.; Validation, O.R.A. and S.V.R.-T.; Visualization, O.R.A.; Writing—original draft, O.R.A.; Writing—review & editing, S.V.R.-T. and A.A.A. All authors have read and agreed to the published version of the manuscript.

**Funding:** This research received no external funding.

**Institutional Review Board Statement:** Not applicable.

**Informed Consent Statement:** Not applicable.

**Data Availability Statement:** The data that support the findings of this study are available at https://figshare.com/s/abe341b40e90c566b061 (accessed on 1 August 2023).

**Conflicts of Interest:** The authors declare no conflict of interest.

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
