# Peer review of "Get Spatial from Non-Spatial Information: Inferring Spatial Information from Textual Descriptions by Conceptual Spaces"

_mathematics, doi:10.3390/math11244917_

Round 1
Reviewer 1 Report
Comments and Suggestions for Authors
1. What is the main question addressed by the research?
This research seeks to represent the concept of space in the real world in a geographic way. In other words, it shows a different way to define different urban areas in a concept model. Researchers did great work and reached an ideal result.
2. Do you consider the topic original or relevant in the field? Does it address a specific gap in the field?
Actually, there is some research on the concept space. It is a very attractive field. For example, “Fine-grained assessment of greenspace satisfaction at regional scale, doi: 10.1016/j.scitotenv.2021.145908” is the NLP method to detect the structure of the green space in Beijing. However, this research realizes a methodological way to conclude the NLP space. There is a specific gap in this field, and this research finely solves this gap.
3. What does it add to the subject area compared with other published material?
For the published material added in comments for editors, the difference is nearly invisible. The contribution of the gap mentioned above is the same as this paper. Though the authors used a different approach, the expression about how to fill the gap is the same. For example, the two articles used the same figure that appears as Fig 2 in this manuscript.
4. What specific improvements should the authors consider regarding the methodology?
The methodology is clear enough for contributable research. For further work, researchers could make a clearer comparison and discussion in interpolation methods and different topics. For example, why some interpolation method is fit for certain topics while others are not?
5. Are the conclusions consistent with the evidence and arguments presented and do they address the main question posed?
Yes, the conclusion and the evidence presented a close connection, and they are addressing the main question.
6. Are the references appropriate?
Yes, the references are appropriate.
7. Please include any additional comments on the tables and figures.
Figures 3, 4, 5, 6, 7 should upgrade the dpi to make it readable.
Author Response
We would like to thank the reviewers for careful and thorough reading of this manuscript and for the thoughtful comments and constructive suggestions. We addressed all the reviewers’ concerns point-by-point. The changed parts of the manuscript are specified in here and in the manuscript. Our responses are as follows (the reviewers’ comments are in italics).
Comment #1: What is the main question addressed by the research? This research seeks to represent the concept of space in the real world in a geographic way. In other words, it shows a different way to define different urban areas in a concept model. Researchers did great work and reached an ideal result.
Response: As the authors of the manuscript, we greatly appreciate your time and effort in reviewing our work. We thank you for positive feedback regarding our manuscript.
Comment #2: Do you consider the topic original or relevant in the field? Does it address a specific gap in the field? Actually, there is some research on the concept space. It is a very attractive field. For example, “Fine-grained assessment of greenspace satisfaction at regional scale, doi: 10.1016/j.scitotenv.2021.145908” is the NLP method to detect the structure of the green space in Beijing. However, this research realizes a methodological way to conclude the NLP space. There is a specific gap in this field, and this research finely solves this gap.
Response: We sincerely thank you for insightful research paper you suggested. In the revised manuscript, we cited the above paper in the literature review. The cited work can be found on Page 4, as:
“Wang, et al. [1] employed content analysis to explore the perception of green spaces in Beijing, China. They developed a structured lexicon based on which landscape features were extracted and fed into machine learning techniques. Their approach relies on prior knowledge about the phenomenon under study.”
Comment #3: What does it add to the subject area compared with other published material? For the published material added in comments for editors, the difference is nearly invisible. The contribution of the gap mentioned above is the same as this paper. Though the authors used a different approach, the expression about how to fill the gap is the same. For example, the two articles used the same figure that appears as Fig 2 in this manuscript.
Response: We thank you for mentioning the paper. In the revised manuscript, we cited the above figure from our latest publication. The main difference between the two approaches is that in the present study we propose both DCS and RCS methods. Then, the results of both methods are compared with those of the LDA method and a random estimator. We discussed the differences between the papers on Page 10, as:
“Abbasi and Alesheikh [52] presented a method similar to DCS with a focus on utilizing it in recommendation systems. They evaluated the results by real users of a recommender system. The main difference between the present study and [52] is that we propose both DCS and RCS and compare their performance against a random estimator, without focusing on recommendation algorithms. In addition, we partitioned the study area into equally-sized cells to compensate for the problem of the density of points in KDE, which is discussed in Results section.”
Comment #4: What specific improvements should the authors consider regarding the methodology? The methodology is clear enough for contributable research. For further work, researchers could make a clearer comparison and discussion in interpolation methods and different topics. For example, why some interpolation method is fit for certain topics while others are not?
Response: We thank you for the constructive suggestion. In the revised manuscript, we provided a discussion over the interpolation methods and different topics on Page 14, as:
“For instance, Topic #D4 contains general features that can be seen in any neighborhood in the study area. This is in contrast with Topics #D1 and #D2 where their associated words is associated with northern and central neighborhoods, respectively.”
and:
“Topic #W1 is related to museums and historical sites, such as palaces, which in turn relate mostly to central neighborhoods and some of the northern neighborhoods.”
and on Page 13, as:
“This is due to the fact that, according to Equation 2, KDE averages the probability distributions over sample points, compensating the role of single points with high values. How-ever, in IDW, single points can have significant effect on the resulting surface.”
Comment #5: Are the conclusions consistent with the evidence and arguments presented and do they address the main question posed? Yes, the conclusion and the evidence presented a close connection, and they are addressing the main question.
Response: We sincerely thank you for your positive feedback regarding our conclusions.
Comment #6: Are the references appropriate? Yes, the references are appropriate.
Response: We thank you again for reviewing our manuscript and appreciate your time and efforts.
Comment #7: Please include any additional comments on the tables and figures. Figures 3, 4, 5, 6, 7 should upgrade the dpi to make it readable.
Response: We thank you for noticing us about the quality of figures. In the revision, we provided high-resolution images to the editorial office.
Reviewer 2 Report
Comments and Suggestions for Authors
This work is interesting and the manuscript is well written. The novelty, i.e., extract spatial information from non-spatial data, is demonstrated by the experiments and discussions. In my opinion, this is a good work which can be accepted after minor revision. Several advices are shown as follows:
1. In the first paragraph of Introduction section, the authors want to deacrib the meaning of geographic information service. I think several new references about geographic information service should be added, such as https://doi.org/10.1111/tgis.12904,https://doi.org/10.1016/j.ecolind.2023.111154.
2. Despite advantages, the limitations, as well as the future work(s), should be analyzed and given in the discussion and conclusion sections.
Author Response
We would like to thank the reviewers for careful and thorough reading of this manuscript and for the thoughtful comments and constructive suggestions. We addressed all the reviewers’ concerns point-by-point. The changed parts of the manuscript are specified in here and in the manuscript. Our responses are as follows (the reviewers’ comments are in italics).
Comment:
This work is interesting and the manuscript is well written. The novelty, i.e., extract spatial information from non-spatial data, is demonstrated by the experiments and discussions. In my opinion, this is a good work which can be accepted after minor revision. Several advices are shown as follows:
Comment 1: In the first paragraph of Introduction section, the authors want to describe the meaning of geographic information service. I think several new references about geographic information service should be added, such as https://doi.org/10.1111/tgis.12904, https://doi.org/10.1016/j.ecolind.2023.111154.
Response: We sincerely thank the reviewer for their careful evaluation of our manuscript. Following the valuable suggestion by the reviewer, we cited the above articles in our introduction (Page 1).
Comment 2: Despite advantages, the limitations, as well as the future work(s), should be analyzed and given in the discussion and conclusion sections.
Response: In the revised manuscript, we provided a discussion on the limitations and future works at the end of the conclusions on Page 18, as:
“The identification of geo-indicative topics in our paper is based on interpreting the entropy values. This interpretation is manually done by comparing the entropies among different topics. As a suggestion, future studies can focus on defining a baseline by which a topic is considered geo-indicative. In addition, various operations can be defined over the conceptual spaces, which makes logical inferences possible. For example, the intersection of two concepts (i.e., regions or directions) may lead to the construction of a new meaningful concept. By studying such operations, the analytical capabilities of the pro-posed approaches can be improved.”
Reviewer 3 Report
Comments and Suggestions for Authors
Dear Authors, I thoroughly review your manuscript. it is really interesting and I would like to suggest you some minor changes for your onward process.
Figure.3, 4,5,6,7. North arrow will place at the right top instead of the left corner top and adjust the legend and scale accordingly.
Figure.5. 6.7. Text coordinates and scale is not clearly visible. Enhanced the visibility or increase the DPI.

Author Response
Reviewer #3
Dear Authors, I thoroughly review your manuscript. it is really interesting and I would like to suggest you some minor changes for your onward process:
Comment 1: Figure.3, 4,5,6,7. North arrow will place at the right top instead of the left corner top and adjust the legend and scale accordingly.
Response: We sincerely thank the reviewer for their careful evaluation of our manuscript. Following the suggestion by the reviewer, we changed the layout of our maps. The north arrow is now at the right corner. The legends and the scale bars are now adjusted as much as possible.
Comment 2: Figure.5. 6.7. Text coordinates and scale is not clearly visible. Enhanced the visibility or increase the DPI.
Response: We thank you again for the positive feedback on our manuscript. In the revision, we increased the size of the scale bars. We also uploaded high-resolution (300 dpi) figures to the submission system so that they can be used in the production stage.